# A CNN-Based Length-Aware Cascade Road Damage Detection Approach

**DOI:** 10.3390/s21030689

**Published:** 2021-01-20

**Authors:** Huiqing Xu, Bin Chen, Jian Qin

**Affiliations:** 1Chengdu Institute of Computer Applications, Chinese Academy of Sciences, Chengdu 610041, China; huazhekanzhai@163.com; 2University of Chinese Academy of Sciences, Beijing 100049, China; 3Harbin Institute of Technology, Shenzhen 518055, China; 4Guangzhou Electronic Technology Co., Ltd., Chinese Academy of Sciences, Guangzhou 510070, China; qinjian@giet.ac.cn; 5Guangzhou Institute of Electronic Technology, Chinese Academy of Sciences, Guangzhou 510070, China

**Keywords:** deep convolutional neural network (CNN), road damage detection, abnormal geometric properties, length-aware, multi-scale attention

## Abstract

Accurate and robust detection of road damage is essential for public transportation safety. Currently, deep convolutional neural networks (CNNs)-based road damage detection algorithms to localize and classify damage with a bounding box have achieved remarkable progress. However, research in this field fails to take into account two key characteristics of road damage: weak semantic information and abnormal geometric properties, resulting in inappropriate feature representation and suboptimal detection results. To boost the performance, we propose a CNN-based cascaded damage detection network, called CrdNet. The proposed model has three parts: (1) We introduce a novel backbone network, named LrNet, that reuses low-level features and mixes suitable range dependency features to learn high-to-low level feature fusions for road damage weak semantic information representation. (2) We apply multi-scale and multiple aspect ratios anchor mechanism to generate high-quality positive samples regarding the damage with abnormal geometric properties for network training. (3) We designed an adaptive proposal assignment strategy and performed cascade predictions on corresponding branches that can establish different range dependencies. The experiments show that the proposed method achieves mean average precision (mAP) of 90.92% on a collected road damage dataset, demonstrating the good performance and robustness of the model.

## 1. Introduction

Currently, due to the diversity of road damage and complex backgrounds, road damage detection is mainly performed manually. However, this process is time-consuming, tedious, subjective, and inaccurate. Multiclass road damage inspection (MRDI), which simultaneously localizes and classifies various types of damage (e.g., cracks, repairs, potholes, crazing, etc.) within high-resolution road images, has become possible due to the increasing imaging quality and development of computer vision technologies. Accurate MRDI is capable of reducing the workload and cost of road maintenance, eliminating the potential threat to traffic safety and improving road service quality.

In recent years, the fast development of convolutional neural networks (CNNs) [1] has greatly raised the bar of computer vision technologies such as image classification [2], semantic segmentation [3], person reidentification [4], object detection [5], etc. Especially in object detection tasks, CNN-based methods report state-of-the-art results on WiderPerson [6], Kitti object [7], Pascal Visual Object Classes (VOC) [8], Microsoft Common Objects in COntext (MS COCO) [9], and other large-scale natural scene object detection datasets. CNN-based object detection algorithms can be primarily divided into one-stage and two-stage detectors. One-stage detectors, represented by you only look once (YOLO) [10,11,12] and single shot detector (SSD) [13,14], directly localize and classify object instances on feature maps. Two-stage detectors, represented by Faster RCNN [15] and its variants [16,17,18] generate sparse proposals on a feature pyramid network (FPN) [19], transform proposals into fixed-size features with region of interest (ROI) pooling [15] or ROI Align [20], and predict location and classification of proposals over the FPN. For image data, large receptive field features model long-range dependencies [21,22]. Two-stage detectors balance the positive and negative samples in the training process, transform the features of proposals, and determine the receptive field feature by the scale of objects, which obtains better performance. In addition, a variety of backbone network networks such as VGG16 [23], ResNet [2] Inception [24], ResNeXt [25], Hourglass [26], Res2Net [27], and HRNet [28] are applied to object detection tasks to obtain improved performance. Recently, it was shown in [29] that CNN-based object detectors are very effective for MRDI on resource-constrained equipment. A Faster RCNN with a backbone network of ResNet was directly applied to MRDI in [30], adjusting model parameters and adopting common data augmentation, which achieved performance far exceeding traditional computer vision technologies. Unlike the proposal generation method of selective search in Fast RCNN [17], traditional image processing techniques such as dilation, erosion, edge detection, contour detection, and convex hull detection were applied in [31] to generate proposals for road damage. The model fed these proposals into a modified SqueezeNet [32] with a learned dictionary to extract features and train damage classifiers. It was identified in [33] that Faster RCNN with backbone network of Inception achieved the best trade-off of detection accuracy and time with a large of number of experiments.

Different from natural and common industrial scenarios, directly applying existing object detection technologies such as Faster RCNN [15] or improved Faster RCNN [34] to MRDI will obtain suboptimal detection results, largely due to the following factors. First, as shown in Figure 1 the road images used in this work have a resolution of 2048 × 2048, and images are easily disturbed by factors such as water stains, garbage, oil spots, gravel, lane lines, imperfect imaging conditions, and noise, resulting in an extremely complex background. Second, there are multiple scales damages; as shown in Figure 1d, the foregrounds of the two instances of damage are 19 and 0.3% of the entire image, respectively. Third, as illustrated in the top row of Figure 1, road damage can visually be curve (Figure 1 a), bar (Figure 1b), block (Figure 1c), or grid (Figure 1d) low-level semantic feature structures, and the damage consists of weak semantic information. Fourth, damage with abnormal geometric properties illustrated in Figure 1a is presented by elongated damage with an unusual aspect ratio; the longest side reaches 2048 pixels and the shortest side has only 150 pixels. The large receptive field features are helpful in detecting elongated damage. Finally, the proposal of elongated damage in Figure 1a will be mapped to the high level of FPN for ROI pooling or ROI Align under the generic two-stage object detectors. However, the excessive downsampling operation of the high level of FPN leads to information loss of elongated damage and makes the network insensitive to the damage boundary [35]. Therefore, both low-level features such as edges, textures, shapes, etc., and the high-level large receptive field features establishing long-range dependencies from the backbone network are required for MRDI. The anchor mechanism of plain two-stage detectors can hardly generate sufficient positive sample anchors to cover damage with abnormal geometric properties. It is better to perform predictions of ROIs on a high-resolution feature map to avoid information loss caused by excessive downsampling.

Inspired by the above observations, based on the characteristics of road damage and Cascade RCNN [18], CrdNet, was proposed. Limited by the current backbone networks [2,28,36], it is difficult to effectively extract features of damage, so we propose a parallel multi-branch LrNet backbone network to extract more discriminative features. In LrNet, for each parallel branch of the proposed backbone network, low-level feature aggregation module (LFAM) reuses low-level features in every sub-stream. For multiple parallel branches, multi-scale feature aggregation module (MFAM) dynamically fuses features of different scales in a fully connected manner with an attention mechanism. The bottom row of Figure 1 shows heat maps of the high-resolution features extracted by the proposed backbone network, indicating that the high-resolution feature maps can obtain clear, continuous, and salient feature representations. According to the distribution of aspect ratios and length of damage in dataset, a multi-scale and multiple aspect ratios anchor mechanism (MSARM) are introduced to generate sufficient positive anchors. Unlike multi-scale object detection of generic two-stage detectors, the receptive field features are determined by the length of the damage in CrdNet. For road damage with multiple lengths, CrdNet uses a set of three convolutions with increasing dilation rates [37] to construct a parallel three-branch structure for detection of small, medium, and long damage. Then road damage is adaptively cascaded, predicted on corresponding branch. To summarize, there are four main contributions of this work:A CrdNet for precise location and classification of road damage is proposed. This proposed model fully considers weak semantic information and abnormal geometric properties of road damage.LrNet outputs fused diversified feature representation from high-to-low level feature representations, across every stage. The fusion of multi-scale convolutional features can meet the feature requirements for detecting road damage.Considering the abnormal geometric properties of road damage, CrdNet utilizes MSARM to generate high-quality positive samples and predict damage of different lengths on three parallel branches.A full labeled dataset containing seven types of damage and 7282 high-resolution images are constructed for performance evaluation, where 6550 road images are used for training the proposed network, and the remaining images are used for testing. The dataset was manually labeled by human experts and contains seven usual types of road damage: longitudinal crack (LC), transverse crack (TC), repair (R), poor repair (PR), pothole (P), crazing (C), and block repair (BR), which can meet the daily basic needs of MRDI. For the uncommon P and C, we use spatial-level transform to augment these two type of less images, which avoids the imbalance of training data. Extensive comparative experiments between the proposed method and several classical generic object detectors are conduced, and the results demonstrate the effectiveness of the proposed method.

The rest of this paper is organized as follows. The proposed method, including the backbone network of low-level features reuse and low-to-high feature fusions, and length-aware head, is presented in Section 2 in detail. The experimental results are presented in Section 3. Finally, Section 4 summarizes the paper with concluding remarks.

## 2. Proposed Method

### 2.1. Overview of CrdNet

As shown in Figure 2, the proposed framework is based on Cascade RCNN [18], which primarily includes the LrNet backbone network, length-aware head, and cascade detection head. LrNet reuses low-level features, dynamically fuses discriminative receptive field features, and outputs four resolutions feature maps, as shown in Figure 3. CrdNet mixes the four resolution representations of LrNet to avoid loss of information, and outputs one enhanced high-resolution multi-scale feature fusion. A set of 3 × 3 convolutions with increasing dilation rates acts on the enhanced features to construct three kinds of parallel high-resolution length-sensitive feature branches for small, medium, and long damage detection. Proposals are generated by a region proposal network (RPN) [15] in CrdNet with MSARM. CrdNet uses an adaptive ROI assignment function to determine which branch is assigned by length. In the cascade detection head, ROI Align [20] is used to extract features for ROIs and perform three-stage cascade proposal classification and bounding box regression. The rest of this section discusses the process in more detail.

### 2.2. LrNet Backbone Network

In this section, LrNet, a backbone network that extracts diversified feature, is proposed to boost the performance of CrdNet. LrNet is improved from HRNet [28] by adding LFAM and MSAM attention modules. High-resolution feature maps of LrNet present diversified feature representations. As shown in Figure 3, LFAM aggregates feature maps of previous convolution blocks in the sub-stream at every stage, which can retain low-level feature representations. The MSAM in Figure 3 utilizes a multi-scale attention mechanism to fuse features of different scales to extract discriminative features in order to avoid feature redundancy.

#### 2.2.1. Parallel Branches of LrNet

LrNet comprises five stages; the first stage is stem [38] consisting of two stride-2 3 *×* 3 convolutions from ResNet. The stem decreases the resolution of input images to one-quarter. The remaining four stages of LrNet are shown in Figure 3. Each stage contains 6-r sub-streams, where r is the number of stages. The sub-streams in one stage are connected serially, and sub-streams in different stages are exhibited in the form of parallel branches. After the data pass through a single branch sub-stream, the sub-streams between the branches are fully connected to each other, and the output features of the branch sub-streams are mixed with the multi-scale feature attention mechanism. Figure 3 details the process of multi-scale attention features fusion between branches. Taking the fusion of three resolution feature maps as an example, the input feature maps can be expressed as {$R_{r}^{i}$, *r* = 2, 3, 4}. The output feature maps, {$R_{r}^{o}$, *r* = 2, 3, 4}, are three resolutions that are the same as the input and one downsampled extra output feature map $R_{5}^{o}$. The process can be expressed as follows:(1)$$R_{r}^{o}=\Gamma (f_{2}^{r}(R_{2}^{i})+f_{3}^{r}(R_{3}^{i})+f_{4}^{r}(R_{4}^{i})),r=2,3,4$$
(2)$$R_{5}^{o}=\Gamma (f_{5}^{2}(R_{2}^{i})+f_{5}^{3}(R_{3}^{i})+f_{5}^{4}(R_{4}^{i})),r=2,3,4$$
where $f_{n}^{m}$(·) is the feature transform function, dependent on the stage numbers (*m*,*n*) of input and output. If *m* = *n*, $f_{n}^{m}$(·) is a normal 3 × 3 convolution. If *m* < *n*, $f_{n}^{m}$(·) is *n − m* consecutive stride-2 3 *×* 3 convolutions for 2*^n−m^* times downsampling. If *m* > *n*, $f_{n}^{m}$(·) is a bilinear upsampling followed by a 1 *×* 1 convolution for 2*^n−m^* times upsampling and aligning the channels of feature maps. Г(*·*) is the MSAM, which realizes multi-scale attention feature fusion. After the data flow and merge across five stages, the high-resolution feature maps of LrNet retain low-level details and enrich the receptive field information at the same time.

#### 2.2.2. LFAM

Each sub-stream consists of four convolutional feature extraction modules and LFAM. Figure 4a illustrates the proposed LFAM. LFAM first concatenates the output features of the four extraction modules, and then utilizes a 1 × 1 convolution to fuse concatenation. The process is shown in Equation (3):(3)$$F_{blend}=F_{c}^{1\times 1}(\{X_{1},X_{2},X_{3},X_{4},\},\theta_{c})$$
where *F_blend_* is the fusion feature of a sub-stream, and $F_{c}^{1\times 1}$ and *θ_c_* represent the 1 × 1 convolution and its parameters, respectively. The directly connected and aggregated strategy of LFAM avoids the difficulty in backpropagation of gradients in sub-streams. The module can effectively extract and retain low-level features in the sub-streams to achieve diversified feature representation.

#### 2.2.3. MSAM

Recently, many attention modules [39,40,41] were embedded into backbone networks to boost the performance of downstream applications. Particularly, channel attention [39] focuses on the meaningful channels of feature maps, while multi-scale attention [40] focuses on the informative contextual scale features. To further boost the performance of LrNet, MSAM fuses multi-scale features from all sub-streams with multi-scale attention, as shown in Figure 3. The main purpose of MSAM is to adaptively select representative scale features regarding the input multi-scale content. The basic idea is to use the gating mechanism to dynamically select branches carrying information of different scales to the next layer. The gating mechanism is divided into two processes, fusion and selection. Figure 4b shows the fusion process of the three scale feature maps. First, the multi-scale features of the three branches are combined via an element-wise summation:(4)$$S\;=\;S_{1}+S_{2}+S_{3}$$

The embedded channel-wise global statistics information as G∈ℝC is generated from combined features *S* by global average pooling. Here the *c*th global statistical information of *G* is calculated as:(5)Gc =1H×W∑i=1H∑j=1WSc(i,j)

To preserve the channel information of *G*, highlight the representative channel and avoid increasing the complexity of the model, *G* is fed into a fully connected layer with *C* channels. And redundant channels of *G* are reduced with a truncated sigmoid function, which preserves the channel information, highlights the representative channels, and avoids aggravating the complexity of the model. The above process is defined as:(6)H =σ(WC(G))
where *H* is refined *G*, *W_c_* is one fully connected layer with *C* channels, and *σ* denotes the truncated sigmoid function. The output of MSAM is a weighted fusion representation, A∈ℝH×W×C, aggregated via channel-wise soft attention, where each channel of the feature map is a weighted combination over input multi-scale features. The computational process of the *c*th channel is calculated as:(7)Ac=∑i=13ai(c)Si
where *a_i_* denotes the weight of the *c*th channel of feature *i*, calculated by:(8)ai(c)=exp(φic(H))∑j=03exp(φjc(H))
and the weight is determined by the mapping φic based on the representation *H*.

### 2.3. Length-Aware Head

In this section, the length-aware head of CrdNet is introduced in detail. The length of the damage, the longest range between the head and tail, corresponds to the longest side of the bounding box of the damage instance and determines what range dependency is required for detection. The length-aware head is mainly composed of three parallel detection branches of different receptive fields and an ROI assignment training scheme based on the length of ROIs. Different from multi-scale object detection on feature maps {P2, P3, P4, P5, P6} with decreasing resolution of FPN, we construct three high-resolution detection branches with different receptive fields to predict multi-length damage, as illustrated in Figure 2. Then the candidate bounding boxes generated by the RPN are assigned to the corresponding branches for cascading prediction by length.

#### 2.3.1. Length-Aware Branches

The three branches of this part are represented by {P2_1, P2_2, P2_3}. The three branches have a stride of 4, and their receptive fields gradually increase. First, in LrNet, the outputs of the three high-level stages are upsampled and merged to the feature map with stride of 4, and it outputs the multi-scale mixed feature P with a stride of 4. Three convolutions with the same structure, except with dilation rates of 1, 6, and 12, are used to create three feature maps of different receptive fields. The receptive fields of the branches model different range dependency, which can be used to detect three kinds of damage of small, medium, and long lengths. Finally, the ROI adaptive assignment strategy is used to assign ROIs to the corresponding branches, and cascade detection is performed on the specific branches.

#### 2.3.2. Adaptive ROI Assignment Function Based on Length

The generic two-stage object detectors assign ROIs to specific layers of FPN with respect to its scale (the area of the bounding box). Then ROI Pooling [15] or ROI Align [20] is used to extract features of ROIs to make predictions. Specifically, it assigns small-scale ROI to lower levels and vice versa. The feature map to be assigned is determined by using Equation (9) in two-stage detectors:(9)$$k=\left\lfloor k_{0}+\log_{2}\sqrt{wh}/224\right\rfloor$$
where *k*_0_ is 4, and *w* and *h* are the width and height of each ROI, respectively. However, for CrdNet, Equation (9) is not suitable for the following reasons. In light of the scale assignment, it is difficult to effectively use the long-range dependencies. For example, Equation (9) will assign an ROI with a length and width of 2000 and 20 pixels, respectively, to the P3. However, the limited receptive fields of P3 are hardly to establish long-range dependency. Besides, the information loss caused by excessive downsampling in the network induces the elongated type of damage to be invisible. Specifically, some ROIs of elongated damage will be assigned to higher levels of the FPN. However, a high level with severe spatial information loss may only retain less useful features for this type of damage. Finally, the canonical size 224 of ImageNet pretraining used in Equation (9) fails to consider the relative scale variation between damage and background. For example, an ROI with an area of 224^2^, a relatively small object only occupying 1.2% of the input image, is assigned to a relative higher feature P4, which reduces detection performance of small damage. Therefore, a new assignment function of ROIs based on length is defined as:(10)$$k=\left\lfloor \frac{5+e^{-x}}{2(1+e^{-x})}\right\rfloor$$
where *x* is the length of the bounding box. Equation (10) adaptively assign ROIs to different receptive field feature maps by length. Compared with Equation (9), the ROI assignment function proposed in this paper improves the detection performance of small and elongated damage because it adaptively selects the appropriate receptive field features based on the length of the damage.

## 3. Experiments

### 3.1. Experimental Setup

#### 3.1.1. Dataset Description

The dataset used for training and evaluation consists of 7282 high-resolution grayscale images with a resolution of 2048 × 2048, called HL2019. These images were collected by a special inspection vehicle from highways in Guangdong Province, China. The dataset contains seven types of challenging damage, carefully selected and labeled by full-time supervisors according to the basic needs of daily road inspections. Figure 5 illustrates the seven types of damage. The dataset has the following characteristics: the images have extremely complex background; there are huge differences in the shape, color, topological structure, and length of damage in different images; some damages closest to the background has low contrast to the background; there is fine-grained detection of repair and poor repair; the imaging is easily affected by unsatisfactory conditions such as weather, light, overexposure, underexposure, shadows, etc., which makes images prone to poor or uneven illumination and further reduces contrast differences between damage and road. Therefore, automatic detection of damage from images is very challenging. In order to evaluate the robustness of the method for damage detection under imperfect imaging conditions, we divided the test data into light and dark types based on the gray value, and report the experimental results.

In the process of constructing the dataset, we used Rotate, Flip and Transpose from Albumentations [42] to enhance the scarce images of C and P. For LC and TC, the type of damage is determined by the aspect ratio of bounding box. If aspect ratio is greater than 1, it will be labeled as LC. It will be labeled as TC when its aspect ratio is less than 1. Since cracks with an aspect ratio of 1 are relatively rare, it is directly marked as a LC when aspect ratio is 1. PR and R may appear in the form of a cross, and we mark multiple damage instances separately. After the image augmentation, we selected a roughly equal amount of data for each type of damage to avoid imbalance in the categories, and the number of images per category are shown in Table 1. In practice, we randomly selected 6550 of 7282 images as training images and the remaining 732 as testing images to evaluate the performance of the models.

#### 3.1.2. Evaluation Metrics

Average precision (AP) and mean average precision (mAP) are common criteria for object detection tasks. We adopted AP and mAP under an intersection over union (IOU) of 0.5, the same definition as in the PASCAL VOC [8] object detection challenge, as the evaluation metric in our experiments. AP was used to evaluate the performance of the models on single damage, reflecting their detection stability on this type of damage. For multi-class damage detection, mAP is the average value of all types of AP, assessing the performance of the models across all types of damage.

#### 3.1.3. Data Preprocessing

The images used for training and testing were 2048 × 2048 high-resolution images. To directly use full-resolution images as input for training requires huge GPU memory, causing training difficulties. In view of this situation, there are usually two methods: cut the image into suitable small patches or shrink the image. The method of cutting the image seriously destroys the integrity of the damage and causes more false prediction. Also, the prediction results of small patches need to be merged again, which increases the difficulty and time of damage detection. In this work, we resized the image to 1024 × 1024 to preserve the integrity of the damage and reduce the need for hardware in the training phase.

#### 3.1.4. Network Setup

The backbone networks used in our experiments for feature extraction were pretrained on MS COCO [9]. We fine-tuned over the training images of HL2019 during the training procedure, but froze the parameters of the first stage of the backbone networks. Much elongated damage exists in HL2019, and the distribution of aspect ratio and length are shown in Figure 6. The aspect ratio and length of the foreground bounding box are concentrated around 0.1 and 2000 pixels. Based on the distribution of damage aspect ratio and length in HL2019, the RPN in this work uses MSARM to generate anchors with more combinations of scales and aspect ratios. Table 2 shows the anchor mechanisms of FPN and CrdNet.

All of our experiments used the MMDetection [43] object detection toolbox based on PyTorch [44], executed on NVIDIA TITAN RTX with a memory capacity of 24 GB. The stochastic gradient descent (SGD) with momentum of 0.9 was adopted to optimize the network. We used pretrained models, froze the first-stage parameters of the models, set the batch size to 2, and trained 60 epochs. The initial learning rate was set to 0.01, and the learning rate dropped to 0.001 and 0.0001 at 20 and 40 epochs, respectively.

#### 3.1.5. Methods to Compare

To verify the contribution of LrNet, MSARM, and CrdNet to the detection of multiple types of damage, we compared four backbone networks with equivalent parameters under the two-stage detectors of Faster RCNN, Faster RCNN+, Cascade RCNN, Cascade RCNN+, CrdNet−, and CrdNet.

ResNet-101 [2]: ResNet-101 consists of 101 convolutional layers. The residual block with identity mapping is introduced to construct a deeper network, which extracts the large receptive field features to establish long-range dependency.

VoVNetV2-57 [36]: VoVNetV2-57 comprises 57 convolutional layers. This network aggregates low-level feature maps at once to capture diverse receptive field features. Then the channel attention mechanism and residual connection are added to refine the features.

HRNetV2-W32 [28]: The initial channels of HRNetV2-W32 is 32. Convolutional information streams of high to low resolution are connected in parallel, and convolutional feature maps of different resolutions are repeatedly merged in a fully connected manner to obtain rich semantic and spatially accurate features.

LrNet: The proposed backbone network is described in Section 2.1.

Faster RCNN [15]: A classic two-stage object detector. We used FPN to generate proposals on feature maps of different scales for damage detection.

Faster RCNN+: Faster RCNN uses MSARM.

Cascade RCNN [18]: A multi-stage object detection architecture based on Faster RCNN that effectively boosts the detection performance of the model by multi-stage progressive prediction refinement and adaptive disposing training distribution.

Cascade RCNN+: Cascade RCNN uses MSARM.

CrdNet−: The damage detection method is introduced in Section 2 without MSARM.

CrdNet: The method is described in Section 2.1.

### 3.2. Results and Analysis

Table 3 presents the results of different methods on the HL2019 test set. The following can be observed:(1)For light and dark images, the mAP_B and mAP_D of all experiments are very similar, indicating that CNNs and two-stage detectors can effectively address the illumination problem of the images and achieve robust damage detection under imperfect imaging conditions. The classic Faster RCNN has the worst performance, especially for elongated types (LC, TC, and PR) and small object types (P), but it effectively detects block repair with regular shapes. Under the same backbone network, Cascade RCNN performs better than Faster RCNN for all categories. The mAP values of Faster RCNN+ and Cascade RCNN+ are both higher than those of Faster RCNN and Cascade RCNN, since MSARM generates high-quality positive samples to cover elongated damage. For CrdNet− without MSARM, the performance over Cascade RCNN+ shows that the information loss of high-level FPN greatly harms the performance.(2)The backbone networks, which extract diversified feature, are very important for damage detection. ResNet-101, which only extracts features establishing long-range dependency, do not perform well under four two-stage detectors. VoVNetV2-57, based on ResNet-101, which aggregates low-level features and extracts diversified feature, boosts detection performance. HRNetV2-W32, which repeatedly integrates feature maps with different resolutions, far exceeds the performance of the previous two backbone networks. Under the four detectors, LrNet perform better than the above three backbone networks with equivalent calculations and parameters. Roughly speaking, LrNet has the merits of ResNet-101, Vovnet57, and HRNetV2-W32: it extracts large receptive field features to establish long-range dependency, effectively retains the low-level features extracted by CNNs and integrates suitable receptive field features with the use of multi-scale attention mechanism. The experimental results show that the above three aspects of the backbone network are important to improve damage detection.(3)Finally, we compared the proposed CrdNet with the baseline. Obviously, compared with the other three two-stage detectors, CrdNet consistently performed better with the four backbone networks. This result demonstrates that prediction on high-resolution feature maps based on the length of damage is superior to prediction on multi-level feature maps of FPN. Under the same baseline backbone network LrNet, the mAP values of faster RCNN+ and Cascade RCNN+ are 80.08 and 82.09%, respectively. The mAP of CrdNet is 90.92%, 9.84% higher than faster RCNN+ and 8.83% higher than Cascade RCNN+. Cascade RCNN+ is a detector based on cascade prediction. But our method can adaptively perform cascade prediction on high-resolution feature maps, which effectively reserves information for elongated and small damages and can see the entire extent of damage by large receptive field features. The benefit of CrdNet over the baseline is that the detection results are highly accurate, and the detection results on HL 2019 are shown in Section 3.4.

### 3.3. Detailed Analysis of Proposed Method

Comparisons of the main results of different methods are presented in the above section. In this section, we perform a detailed analysis of LrNet, length-aware head, and cascade detection head to further understand how they work.

#### 3.3.1. Ablation Study

Firstly, we investigate the effects of LFAM, MSAM, Multiple Branches and Cascade head in CrdNet. LFAM and MSAM indicate low-level features resusing and scale attention modules respectively, Multiple Branches denote high resolution road damage prediction branches, and Cascade means performing cascade prediction on the specified branch. From Table 4, we can learn that the LFAM from LrNet backbone improve the mAP by 6.37%, and the MSAM contributes to a further 5.59% improvement. The cascade head leads to a gain of 7.07%. The component-wise analysis of CrdNet will be discussed in the following sub-sections.

#### 3.3.2. LrNet

We first examine the feature representation ability of LFAM and MSAM in LrNet. The results in Table 5 show that LrNet improves damage detection performance. LFAM effectively reuses low-level features, alleviates information loss caused by network depth, and extracts diversified feature to meet the needs of low-level semantic features for damage detection. At the same time, for the network of gradient backpropagation, gradient directly acts on the low layers by LFAM, which improves the feature representation ability of the low layers in the LrNet. MSAM further improves the performance. In the process of feature fusion between multiple branches, MSAM considers the contribution of different receptive field features to the fusion features, and extracts discriminative features.

#### 3.3.3. Length-Aware Branches and Length-Adaptive ROIs Assignment Function

Unlike FPN, when CrdNet uses Equation (9), *k*_0_ is set to 1, the maximum and minimum values of *k* are set to 3 and 1 respectively, and ROIs are predicted on three length-aware branches. Table 6 shows the results of single branch and multi-branch by applying Equations (9) or (10). Compared with the single branch, multiple branches achieve an increase over the baseline. However, multi-branch with Equation (9), only achieves slight improvement over the baseline. This is because Equation (9) assigns most of the ROIs to the P2_3 branch with a larger receptive field for prediction, and the multi-branch almost degenerates into a single branch. Equation (10) takes the length of ROIs into account, and adaptively assigns ROIs to specific branches, which is 8.71% higher than Equation (9). This means that multiple branches provide different receptive field features to detect ROIs. Equation (10) assigns ROIs to appropriate branches with respect to the ROI length to avoid degradation of multiple branches.

#### 3.3.4. Cascade Detection Head

We studied the choice of the number of cascade stages in CrdNet. Table 7 shows the cascade results of CrdNet using one to four stages. The results in Table 7 demonstrate that cascade detection head consistently boosts the performance over the single-stage method (baseline) with 2.41 to 7.07% mAP increase. However, the continuous addition of cascade detection stages does not bring any more improvement. It can be seen from Table 7 that the performance of four-stage cascade detection is degenerated. Thus, we chose three-stage cascade detection as the default setting from the view of model complexity and performance.

### 3.4. Qualitative Analysis of CrdNet

The previous experiments show the effectiveness of the proposed method on road damage detection. In this section, we qualitatively analyze the features extracted by LrNet, the anchors generated by MSARM, and the prediction results of CrdNet to explain why CrdNet yields good detection performance.

#### 3.4.1. Visualization of High-Resolution Feature Maps of Backbone Networks

CrdNet predicts results on feature maps based on the second-stage output of LrNet. Thus, we visualized the second-stage feature maps of the four backbone networks. As shown in Figure 7, for two images under different illumination, LrNet is able to effectively establish the long-range dependency of the entire extent of damage, produce a continuous response to the foreground area, and suppress the attention on the background area. However, the third, fourth, and fifth columns of Figure 7 show that HRNetV2-W32 [28], VoVNetV2-57 [36], and ResNet-101 [2] only respond to the local area of damage, and it is difficult for them to establish a continuous feature response to the foreground area. Compared with the features extracted by LrNet, the other three backbone networks have better response to the background area, and the extracted features have more noise. On the basis of HRNetV2-W32, LrNet extracts continuous, clear, and salient foreground feature by reusing the low-level features and multi-scale attention feature fusion mechanism of the multiple branches.

#### 3.4.2. MSARM

The anchor-based two-stage detector is a kind of heuristic algorithm. If the scale and aspect ratio of the RPN do not fit the distribution of the training data, it leads to poor performance. Figure 8 illustrates the anchors generated by RPN under the two anchor mechanisms. When IOU is 0.7, the positive samples generated by the two anchor mechanisms have a significant drop. To increase the recall of the models, we chose 0.5 as the IOU threshold for RPN during training and testing. The proposed MSARM generated more and higher-quality positive sample anchors than FPN under different IOUs. This means that the anchors generated by MSARM effectively cover damage with abnormal geometric properties, which avoids the CrdNet only learn the background area of images and improves the damage detection results.

#### 3.4.3. Analysis Results of CrdNet

Figure 9 shows a few samples of CrdNet detection results from the HL2019 test set. For the special elongated damage, this method accurately localizes and classifies the damage, as shown in the first row of Figure 9. Figure 9b,d show that the proposed method is able to tackle illumination problems of images. This method can not only localize two intersecting damage instances, but also provide fine-grained prediction of two similar damage instances, and the detection results of two intersecting damage instances are shown in Figure 9d. The visualization results in Figure 9e show that this method can successfully detect small pothole-type damage. Finally, Figure 9e,f show that this method predicts accurate boundaries of block-type damage based on the rich contextual information extracted by LrNet. The good detection performance of CrdNet is largely attributed to reusing low-level features of LrNet, extracting diversified feature, balancing multi-branch multi-scale feature fusion with multi-scale attention, MSARM, and cascaded predictions on the branches that establish different range dependencies.

The third row of Figure 9 shows several detection failures of CrdNet. First, the dashed yellow bounding box in Figure 9g represents false negative crack detection, largely because the backbone network is unable to effectively extract discriminative features of this subtle damage, or the concerned training images within the training set are scarce. Second, CrdNet misreports poor repair as repair, as shown in Figure 9h. After analyzing this false positive, it was found that this was due to duplicate annotations on the same image. The supervisors labeled this type of damage as both repair and poor repair. Our model confused the detection of this kind of damage that is difficult for humans to define. Finally, Figure 9i shows that we have a wrong prediction of block repair. Block repair generally has the traits of relatively regular shape, simple internal topological structure, a large gap with the background and several corners, and the corners are the key features in this damage. The main reason for the false positive in Figure 9i is that the backbone network failed to actively pay more attention to the key features of the damage in the process of extracting features. In future work, it will be necessary to explicitly add prior knowledge of the damage in the feature extraction process to address this failure prediction.

## 4. Conclusions

In this paper, we propose CrdNet, a road damage detection method based on the weak semantic information and abnormal geometric properties of road damage. This method primarily consists of LrNet backbone network, MSARM, and length-aware and cascaded detection heads. LrNet reuses low-level features and fuses discriminative receptive field features to extract features matching the geometric and semantic properties of the damage. MSARM generates high-quality positive samples with respect to the distribution of geometric properties of damage, and addresses the constraint of general two-stage detectors only generating a limited number of anchors. Finally, cascade predictions were adaptively performed on different branches based on the length of the damage. The results showed that our method is superior to previous two-stage detectors in terms of accuracy and effectiveness. Nevertheless, the following issues are worth considering in future work:(1)Predicting damage instances with a bounding box is a relatively rough location method. It is not able to pick up more concrete geometric properties of damage, which makes it difficult to provide more useful damage indicators for assessment in practice.(2)To minimize the information loss of shrinking high-resolution images, the resized image and patches of the image should both be considered as the input of CNNs.(3)CNNs that extract features in an automatic learning fashion may neglect the key features of certain damage, leading to failure to predict the damage. In the future, explicitly adding prior knowledge of road damage to the feature extraction process of CNNs addresses the failure prediction.

## Figures and Tables

**Figure 1 sensors-21-00689-f001:**
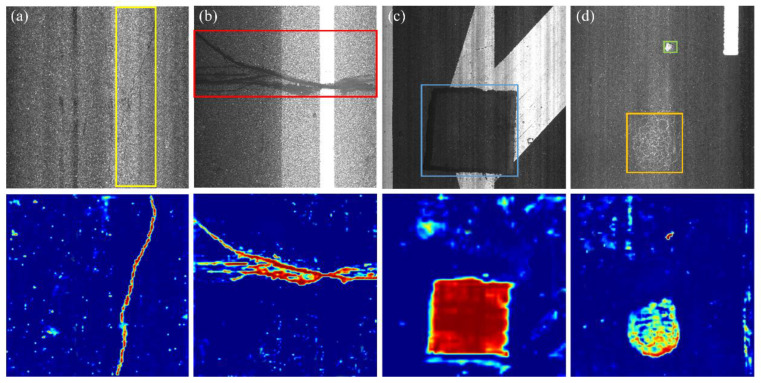
Five common types of damage. (**a**) Curvilinear damage. (**b**) Bar-like damage. (**c**) Block-like damage. (**d**)Grid-like and small damages. Damage is extremely susceptible to disturbance by complex backgrounds and unsatisfactory imaging conditions. Second row shows visualizations of the highest resolution feature map by the proposed backbone network.

**Figure 2 sensors-21-00689-f002:**
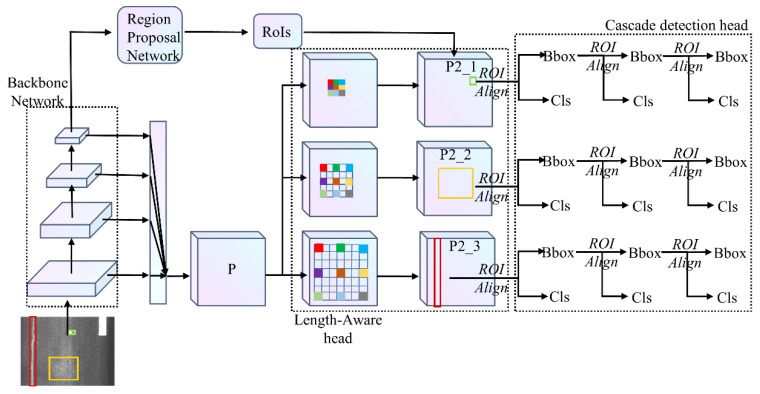
Overall architecture of CrdNet. High-resolution feature map P (stride of 4) denotes fused feature map of four stage output from backbone network. Three branches of length-aware head extract receptive fields to establish different range dependencies. Cascade detection head adaptively cascades to predict on the corresponding branch with respect to the length of regions of interest (ROIs).

**Figure 3 sensors-21-00689-f003:**
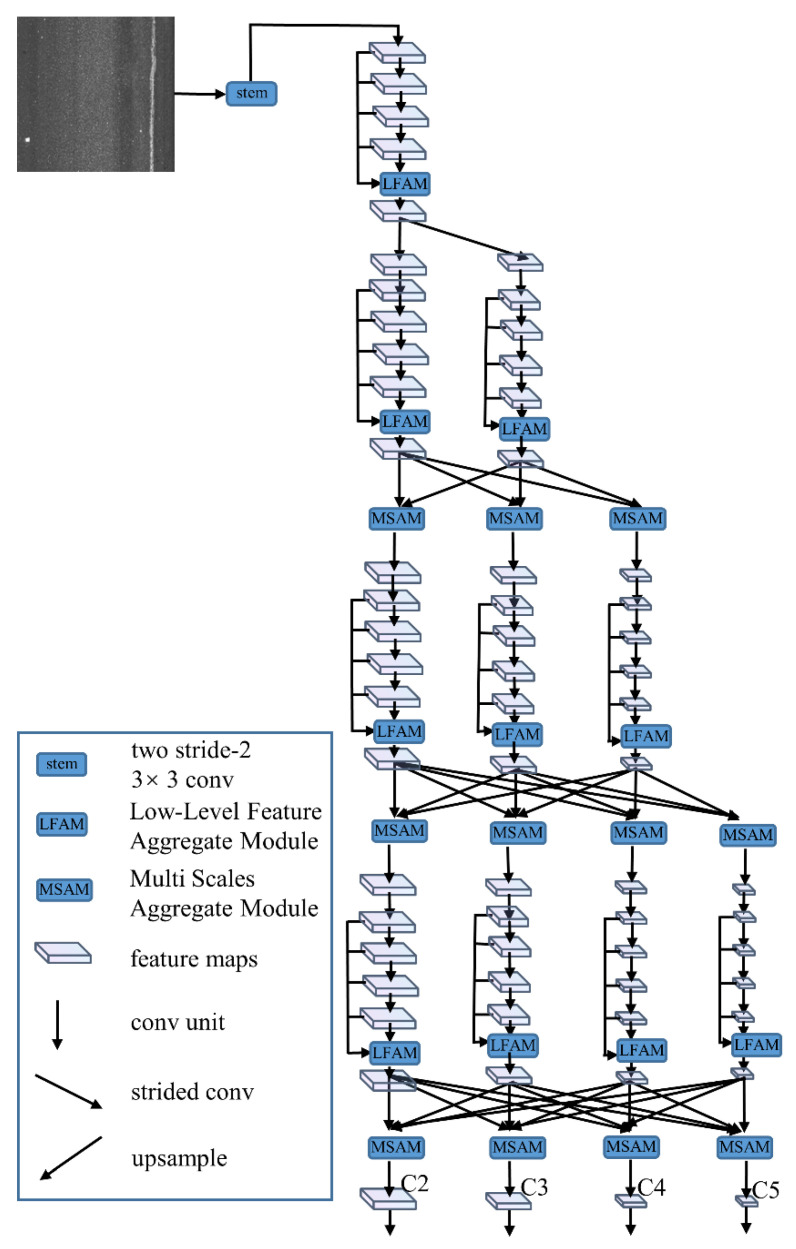
Proposed LrNet, where conv is convolution operation. LrNet uses low-level feature aggregate module (LFAM) to aggregate low-level features of each sub-stream of the parallel branch, then uses multi-scale aggregate module (MSAM) to fuse parallel branch outputs in a fully connected manner. Finally, it outputs 4 kinds of resolution feature maps: C2 (stride of 4), C3 (stride of 8), C4 (stride of 16), and C5 (stride of 32).

**Figure 4 sensors-21-00689-f004:**
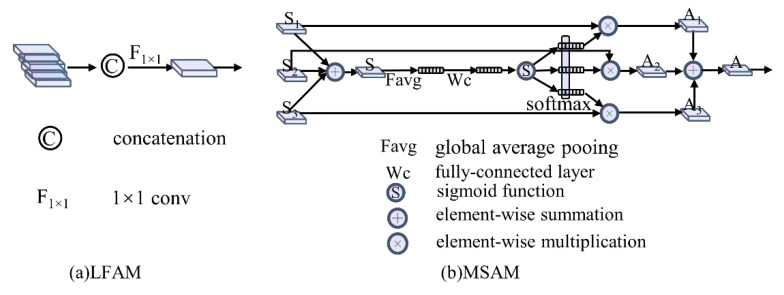
Illustration of LFAM and MSAM.

**Figure 5 sensors-21-00689-f005:**
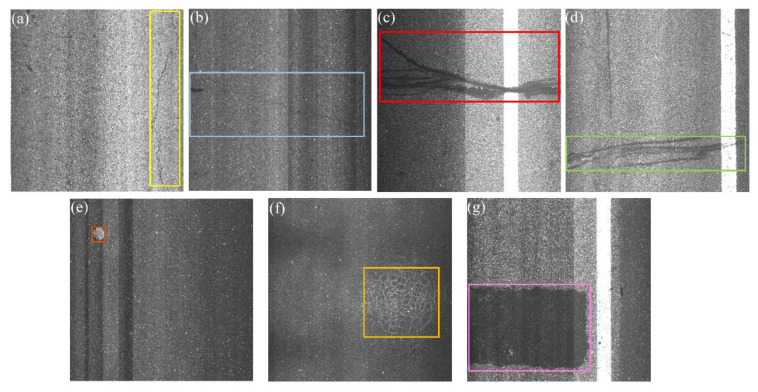
HL2019 dataset. (**a**) Longitudinal crack (LC). (**b**) Transverse crack (TC). (**c**) Repair (P). (**d**) Poor repair (PR). (**e**) Pothole (P). (**f**) Crazing (C). (**g**) Block repair (BR).

**Figure 6 sensors-21-00689-f006:**
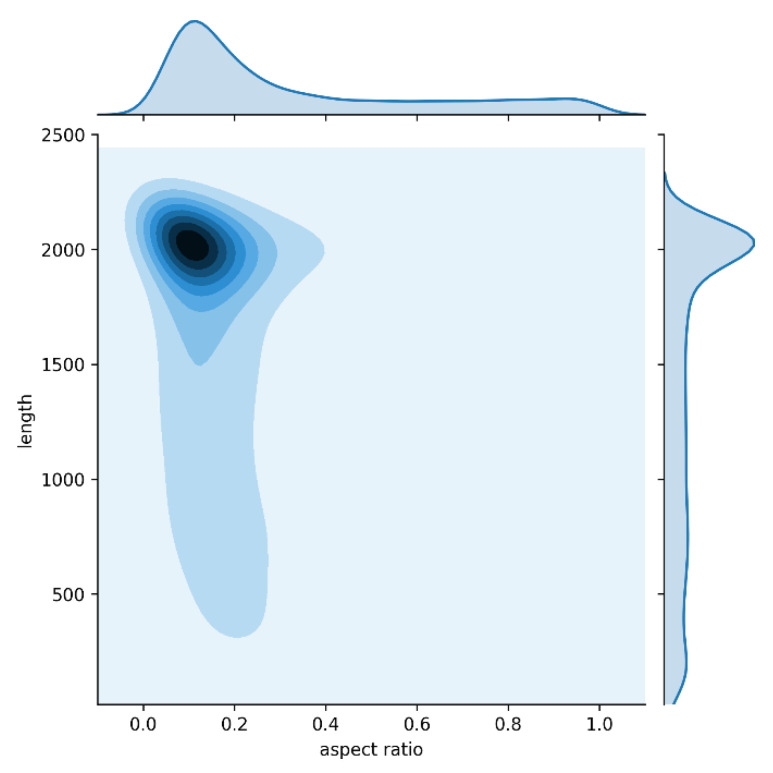
Aspect ratio and length distribution of damage in the dataset. Center of the image is joint distribution of aspect ratio and length. Top and rightmost parts are aspect ratio and length distribution, respectively.

**Figure 7 sensors-21-00689-f007:**
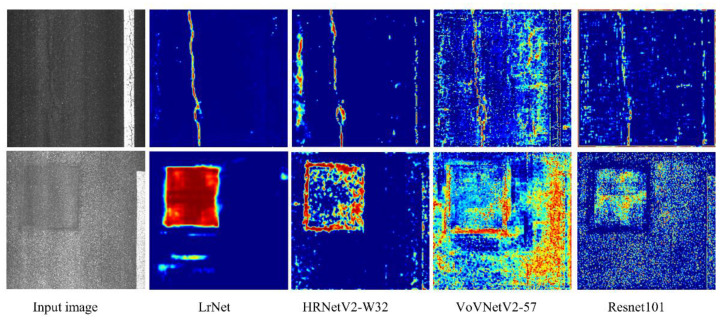
Visualizations of second-stage feature maps of four backbone networks. Red and bright areas denote strong response, blue indicates areas of no concern.

**Figure 8 sensors-21-00689-f008:**
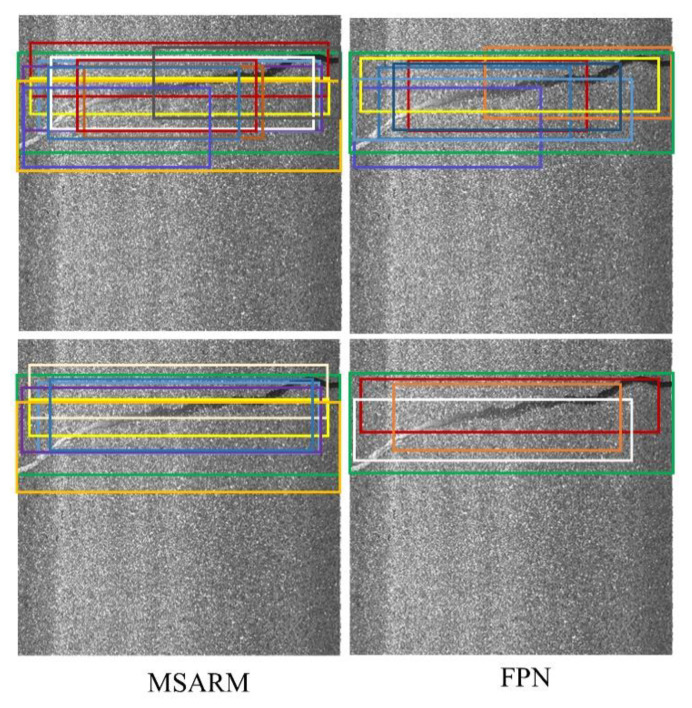
Anchors generated by two anchor mechanisms. Green bounding box indicates ground truth, and other colors indicate generated anchors. Rows 1 and 2 represent anchors matching ground truths with the two mechanisms when IOU is 0.5 and 0.7, respectively.

**Figure 9 sensors-21-00689-f009:**
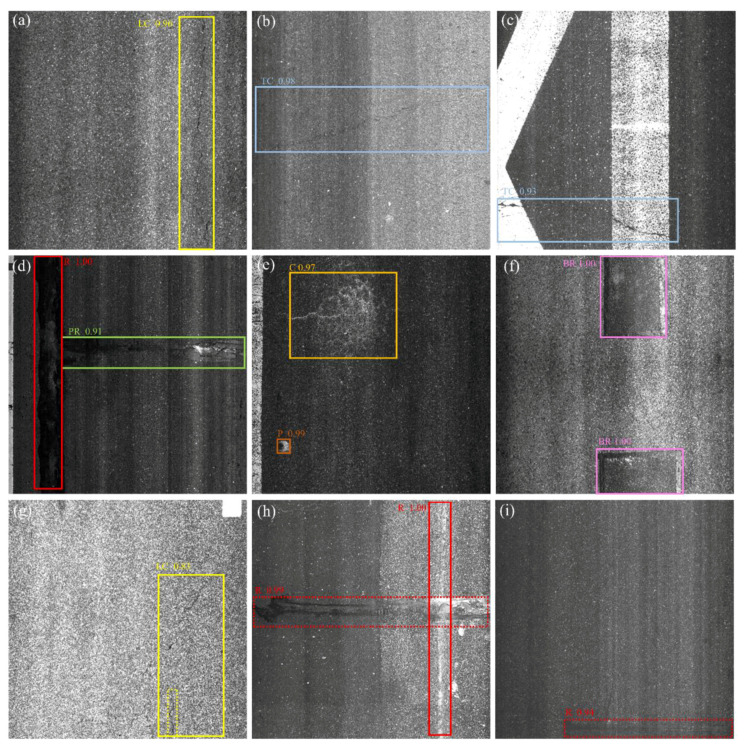
Results of CrdNet HL2019 test set. (**a**–**f**) in first two rows are true positive results of proposed method. (**g**–**i**) in last row represents typical failure cases of CrdNet; dashed box with text on top represents false positive, and dashed box without text represents false negative.

**Table 1 sensors-21-00689-t001:** The number of images per category in HL2019.

Category	Image Number
LC	1054
TC	1047
R	1059
PR	1046
C	1003
P	1007
BR	1066

**Table 2 sensors-21-00689-t002:** Two anchor mechanisms used in training: feature pyramid network (FPN) and multi-scale and multiple aspect ratios mechanism (MSARM).

Method	Scales	Aspect Ratios
FPN	[32, 64, 128, 256, 512]	[0.5, 1.0, 2.0]
MSARM	[32, 48, 64, 96, 128,192, 384, 512, 768]	[0.1, 0.5, 1.0, 2.0, 10]

**Table 3 sensors-21-00689-t003:** Performance comparison of HL2019 test set. mAP_B and mAP_D, mean average precision of light and dark images, respectively. + indicates Faster RCNN and Cascade RCNN use MSARM. − means CrdNet utilizes FPN to generate anchors.

Method		Average Precision (%)									
	Backbone	LC	TC	R	PR	C	P	BR	mAP (%)	mAP_B (%)	mAP_D (%)
FasterRCNN	Resnet101	42.33	47.26	66.32	49.94	59.26	43.67	81.02	55.69	56.23	55.14
CascadeRCNN	Resnet101	45.63	49.52	68.37	53.45	63.52	48.26	83.56	58.90	59.62	58.18
CrdNet −	Resnet101	50.34	51.63	70.28	58.24	67.97	53.29	84.68	62.35	63.21	61.48
Faster RCNN +	Resnet101	44.36	48.33	67.66	52.67	60.12	46.38	81.26	57.25	57.78	56.72
Cascade RCNN +	Resnet101	46.28	49.54	68.14	54.21	64.26	51.29	84.65	59.77	59.92	59.62
CrdNet	Resnet101	51.68	52.42	72.35	60.81	68.33	55.86	85.12	63.80	64.47	63.12
Faster RCNN	VoVNetV2-57	43.56	48.85	67.25	51.26	62.44	46.89	82.05	57.47	58.43	56.51
CascadeRCNN	VoVNetV2-57	46.58	50.67	68.51	53.41	64.26	48.83	83.98	59.46	59.96	58.97
CrdNet −	VoVNetV2-57	51.29	52.04	71.56	60.08	68.26	54.18	84.97	63.20	63.36	63.03
Faster RCNN +	VoVNetV2-57	44.65	49.27	69.29	52.33	63.48	48.39	83.11	58.65	59.06	58.23
Cascade RCNN +	VoVNetV2-57	48.12	51.24	70.08	54.29	65.26	50.04	84.96	60.57	61.06	60.08
CrdNet	VoVNetV2-57	52.58	52.47	71.53	61.39	69.25	57.42	84.51	64.16	64.91	63.42
Faster RCNN	HRNetV2-W32	49.36	56.43	76.54	65.89	67.78	61.96	88.41	66.62	66.93	66.31
Cascade RCNN	HRNetV2-W32	52.43	59.42	79.74	67.34	69.24	64.20	93.31	67.95	68.47	67.44
CrdNet −	HRNetV2-W32	53.58	60.74	80.63	69.74	71.24	66.95	92.46	70.76	71.04	70.49
Faster RCNN +	HRNetV2-W32	50.63	57.58	77.93	66.23	68.72	62.30	88.93	67.47	68.14	66.81
Cascade RCNN +	HRNetV2-W32	53.67	61.52	80.35	68.76	70.94	66.83	93.68	70.82	71.68	69.96
CrdNet	HRNetV2-W32	55.64	61.97	80.86	71.38	72.99	69.61	93.42	72.27	73.18	71.35
Faster RCNN	LrNet	62.53	71.39	82.59	77.37	87.06	73.12	94.59	78.39	78.63	78.15
Cascade RCNN	LrNet	63.94	72.42	84.52	78.52	88.42	75.94	95.24	79.86	80.47	79.24
CrdNet −	LrNet	79.36	79.89	86.77	84.35	91.68	85.16	95.81	86.18	86.98	85.39
Faster RCNN +	LrNet	70.28	73.26	84.56	79.24	89.45	75.94	94.85	81.08	82.02	80.15
Cascade RCNN +	LrNet	71.25	74.84	85.27	80.76	90.63	76.91	94.99	82.09	82.80	81.39
CrdNet	LrNet	88.34	89.25	90.94	89.92	91.83	90.54	95.62	90.92	91.07	90.77

**Table 4 sensors-21-00689-t004:** Effects of each component in CrdNet. Results are reported on HL2019 test set.

Multiple Branches	LFAM	MSAM	Cascade	mAP (%)	mAP_B (%)	mAP_D (%)
√				71.89	72.35	71.43
√	√			78.26	78.65	77.87
√	√	√		83.85	84.04	83.66
√	√	√	√	90.92	91.07	90.77

**Table 5 sensors-21-00689-t005:** Effects of LFAM and MSAM on LrNet under CrdNet. LrNet is improved by adding LFAM and MSAM to HRNetV2-W32. + denotes adding the corresponding component to HRNetV2-W32.

Backbone	mAP (%)	mAP_B (%)	mAP_D (%)
HRNetV2-W32 [28]	72.27	73.18	71.35
+LFAM	83.60	83.86	83.34
+MSAM	90.92	91.07	90.77

**Table 6 sensors-21-00689-t006:** Results of length-aware head based on LrNet. – indicates single branch.

Method	No. of Branches	mAP (%)	mAP_B (%)	mAP_D (%)
CrdNet (baseline)	–	80.23	80.36	80.10
CrdNet	3 branches (Equation (9) [19])	82.21	82.93	81.49
CrdNet	3 branches (Equation (10))	90.92	91.07	90.77

**Table 7 sensors-21-00689-t007:** Results of CrdNet on HL2019 test set using different numbers of cascade stages on LrNet.

Stage	mAP (%)	mAP_B (%)	mAP_D (%)
1	83.85	84.04	83.66
2	86.26	87.15	85.37
3	90.92	91.07	90.77
4	90.38	90.67	90.09

## Data Availability

Not applicable.

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
