# Peer review of "A CNN-Based Length-Aware Cascade Road Damage Detection Approach"

_sensors, 2021, doi:10.3390/s21030689_

Round 1

Reviewer 1 Report

I think that this MS presents a comprehensive work about the Road Damage Detection. 

Author Response

Thank you very much for your affirmation and guidance of our works.

Reviewer 2 Report

By using deep convolutional neural networks, they tried to detect the road surface damages. In particular, the proposed method is based on the object detection with bounding boxes. Here, I have some questions.

1) Page 3 

A full labeled dataset containing seven types of damage and 7282 high-resolution images are constructed for performance evaluation, where 6550 road images are used for training the proposed network, and the remaining images are used for testing. The dataset was manually labeled by human experts and contains seven usual types of road damage: longitudinal crack(LC), transverse crack(TC), repair(R), poor repair(PR), pothole(P), crazing(C), and block repair(BR), which can meet the daily basic needs of MRDI. 

  • A total of dataset is 7,282 having seven classes. How many numbers of data are each class composed of? 
  • Is there any augmentation?
  • It there any imbalanced? 
  • The dataset was manually labeled by human experts. I think that there are ambiguous data during annotation process. For example, are there any cracks between the longitudinal crack and the transverse crack? If these cracks exist, it would be nice if it was described how to solve them.

2) 3.1.5. Methods to Compare

  • Please clarify your sentences. The sentences and phrases are mixed.
  • "ResNet-101[2]: ResNet consists of 101 convolutional layers." -> The resnet has various versions with different layers. Your sentence is said to have only 101 layers. 

3) Page 10

We froze the parameters of the first stage of the backbone networks and then fine-tuned over the training images of HL2019 during the training procedure.

  • I don't understand this sentence. You mean there are two training steps. At the first step, the backbone was frozen. After that, at the second step, the backbone was also fine-tuned. Is this right? Then, what is another difference between first and second steps?

4) Ablation Study

  • I think the performance difference according to the backbone or network is too great. Because the number of test images is small, this results can be reproduced. 
  • If you can, it would be nice to add an ablation study that can increase the reliability of the current results.

Author Response

Point 1: Page 3 

A full labeled dataset containing seven types of damage and 7282 high-resolution images are constructed for performance evaluation, where 6550 road images are used for training the proposed network, and the remaining images are used for testing. The dataset was manually labeled by human experts and contains seven usual types of road damage: longitudinal crack(LC), transverse crack(TC), repair(R), poor repair(PR), pothole(P), crazing(C), and block repair(BR), which can meet the daily basic needs of MRDI. 

  • A total of dataset is 7,282 having seven classes. How many numbers of data are each class composed of? 
  • Is there any augmentation?
  • It there any imbalanced? 

The dataset was manually labeled by human experts. I think that there are ambiguous data during annotation process. For example, are there any cracks between the longitudinal crack and the transverse crack? If these cracks exist, it would be nice if it was described how to solve them.

Response 1:

  • The number of images of longitudinal crack, transverse crack, repair, poor repair, pothole, crazing, and block repair are 1054,1047,1059,1046,1003,1007, and 1066 respectively. Talble 1 in the page 9 lists the details of HL2019.
  • Yes, we adopted some augmentation for pothole and crazing, because they uncommon. The augmentation is just spatial-level transform such as Rotate, Flip and Transpose from Albumentations. Introduction in page 3 briefly describe the augmentation and page 9 details the used methods and the final form the data set.
  • Yes, the imbalance of categories exists. Mainly the pothole and crazing are less. And we augmented these two types images to avoid poor performance.

Point 2: 3.1.5. Methods to Compare

  • Please clarify your sentences. The sentences and phrases are mixed.
  • "ResNet-101[2]: ResNet consists of 101 convolutional layers." -> The resnet has various versions with different layers. Your sentence is said to have only 101 layers. 

Response 2:

  • I have been changed all the phrases into sentences to make them more precise. Please see the change in new 3.1.5. section.
  • We are not saying that resnet has only101 layers. We are referring to the ResNet-101 has 101 convolutional layers. We have revise the missing in page 11 of the new paper.

Point 3: Page 10

We froze the parameters of the first stage of the backbone networks and then fine-tuned over the training images of HL2019 during the training procedure.

  • I don't understand this sentence. You mean there are two training steps. At the first step, the backbone was frozen. After that, at the second step, the backbone was also fine-tuned. Is this right? Then, what is another difference between first and second steps?

Response 3:

  • We have no two training steps. We fine-tune over the training images of HL2019 during the training procedure, but freeze the parameters of the first stage of the backbone networks. 

Point 4: Ablation Study 

  • I think the performance difference according to the backbone or network is too great. Because the number of test images is small, this results can be reproduced. 
  • If you can, it would be nice to add an ablation study that can increase the reliability of the current results.

Response 4:

  • We plan to continuously expand the data set based on the current results. We further verify the effectiveness of the proposed method on a larger data set.
  • We conduct ablation experiments on the four key components of CrdNet: LFAM, MSAM, Multiple Branches and Cascade head. We add the experiment content to the 3.3.1. section.

Reviewer 3 Report

The authors presented a a road damage detection method accounting the
weak semantic information and abnormal geometric properties of road damage, usually not addressed properly which limits detection result.

The paper is generally well written and organized. In addition the results which are quite promising, are summarized and the method compared in terms of performance with other approaches. 

All in all, I believe the paper worthy for publication in the present form.

Author Response

(The authors gave the same response as above.)
